# How the Implementation of BREEAM in Hotels Could Help to Achieve the SDGs

Maria M. Serrano-Baena *, Rafael E. Hidalgo Fernández, Pilar Carranza-Cañadas and Paula Triviño-Tarradas

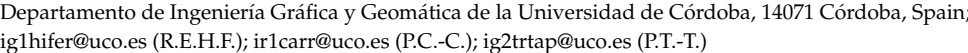

Departamento de Ingeniería Gráfica y Geomática de la Universidad de Córdoba, 14071 Córdoba, Spain;
ig1hifer@uco.es (R.E.H.F.); ir1carr@uco.es (P.C.-C.); ig2trtap@uco.es (P.T.-T.)
* Correspondence: ep2sebam@uco.es

**Abstract:** The 2030 Agenda for Sustainable Development and its 17 Sustainable Development Goals (SDGs) was approved in 2015 by the United Nations. It is a call of action to protect our planet, end poverty and improve the lives and prospects of all. Sustainable development has been fundamental in the tourism and construction sectors in the past few decades. Nowadays, developing countries are leaders in green engineering procedures, and progressively, hotels are including sustainable standards in their designs, architecture and management. In places where tourism is the main contributor to the Gross Domestic Product, the incorporation of energy certifications is crucial. In this context, this article explores the positive implications of the application of the Building Research Establishment Environmental Assessment Method (BREEAM) on hotels in relation to the achievement of SDGs. The study analyses the influence of BREEAM on hotel design using six case studies and examines the sustainable modifications incorporated. Qualitative data were obtained through in-depth interviews and by the analysis of the documentation provided. The results revealed that a BREEAM approach in the initial stage of a project will optimize the sustainability of the hotel and can help with the achievement of several of the SDGs.

**Keywords:** sustainable hotel; hospitality; innovative design; sustainability assessment; green engineering; 2030 Agenda



## 1. Introduction

In 2015, the Heads of State and Government of 193 countries met at the 70th General Assembly of the United Nations and approved the resolution of "The 2030 Agenda for Sustainable Development". This plan establishes 17 Sustainable Development Goals (SDGs) and 169 targets to be achieved by 2030 that are currently considered a global emergency to mitigate and balance the three dimensions of sustainable development: the economic, social and environmental [1]. The EU is playing an active role and will implement the 2030 Agenda domestically and globally in cooperation with partner countries [2]. Furthermore, the New Urban Agenda adopted in October 2016 at the United Nations Conference aims to accelerate the achievement of the SDGs. Particularly, it prioritizes the making of cities safer, more inclusive and sustainable by adopting a smart-city approach [3]. Smart-city models require specific strategies to optimize their resources, reduce its waste and recycle; in this context, the generation of a circular economy plays an important role [4].

As demonstrated in the past few decades, sustainable development has been fundamental in the tourism and construction sectors [5]. Nowadays, developing countries are leaders in green engineering procedures, and progressively, hotels are including sustainable standards in their designs, architecture and management [6]. The World Tourism Organization (UNWTO) has stated that tourism contributes directly or indirectly to the achievement of all the SDGs and is particularly included in objectives 8, 12 and 14.

1. SDG 8. "Promote inclusive and sustainable economic growth, employment and decent work for all" [7]. Travel and tourism maintain a total of 319 million jobs worldwide

and induce 10% of all jobs [6,8]. The contribution of the tourism sector is specified in target 8.9 "By 2030 devise and implement policies to promote sustainable tourism which creates jobs, promotes local culture and products" [1,9].

2. SDG 12. "Ensure sustainable consumption and production patterns" [7]. By adopting responsible consumption and production practices, the tourism sector can significantly accelerate the shift towards a more sustainable planet [7]. The One Planet Sustainable tourism Program aims to improve the impact of tourism on sustainable development through the promotion of responsible consumption and production practices, which use natural resources and generate less waste [10].

3. SDG 14. "Conserve and sustainably use the oceans, seas and marine resources" [7]. It is specified in target 14.7 "By 2030 increase the economic benefits to Small Islands Developing States (SIDS) and Least Developed Countries (LDCs) from the sustainable use of marine resources, including through sustainable management of fisheries, aquaculture and tourism" [1]. Integrated coastal zone management must include tourism development to preserve fragile marine ecosystems and promote a blue economy [7].

*BREEAM and the Sustainable Development Goals*

BREEAM supports the SDGs and notes how it significantly contributes to the achievement of the following goals by applying these methods [11]:

1. SDG 3. Health and Well-being. The building must guarantee minimum requirements for comfort and health.
2. SDG 6. Clean Water and Sanitation. Buildings must install systems that save water as well as monitor their consumption.
3. SDG 7. Affordable and clean energy. It promotes the installation of renewable energy sources and the use of low consumption appliances and lighting.
4. SDG 9. Industry, Innovation and Infrastructure. Rate and promote those projects that create or offer new sustainable solutions.
5. SDG 11. Sustainable Cities and Communities. An urban environment requires different measures to those applied in a single building; therefore, the certificate proposes a tool that evaluates and certifies urbanized spaces.
6. SDG 12. Sustainable Consumption and Production. It requires eco-labels that guarantee the responsible sourcing of materials used on site.
7. SDG 13. Climate Action. Its main objective is to guide the construction sector towards sustainability.
8. SDG 15. Life of Terrestrial ecosystems. It seeks to minimize the damage caused to the environment and its biodiversity when building.

Although the UNWTO has established that tourism contributes to the achievement of the SDGs [2], the hotel industry faces the challenge of determining which of the 17 SDGs and their associated targets are its priority [12]. Hotel guests have radically changed their attitude towards adopting environmentally friendly practices, and, as a result, hotels cannot ignore their environmental and social responsibilities [13–15]. Research on efficient energy-use procedures and interventions based on clients energy-related behavior is crucial for hoteliers to understand what the guests' needs are and how these can be incorporated in the hotel [16].

A sustainable hotel must operate according to the principles of green engineering by implementing waste management systems, recycling and saving water and energy, among other procedures [17]. These approaches must be incorporated from the design stage of the building. A recent study has shown that the application of environmental procedures and their consequent certifications can optimize the image of the firm and its operational performance [18]. The EU Energy Efficiency Building Directive (EPBD) and the UK Climate Change Act 2008 included sustainable buildings on their policy agenda. Since then, a wide variety of tools have been developed to assess and assist construction projects, with the Building Research Establishment Environmental Assessment Method (BREEAM) being among the most successful of them [19]. BREEAM leads the list and, although it was

initially designed to focus primarily on environmental aspects [20,21], in the last decade it has also highlighted social and economic aspects. Despite its global implications, there is not much research on how improving hotel sustainability can help successfully achieve the SDGs of the 2030 Agenda.

This paper studies the application of the BREEAM energy certificate in hotel design. It shows how the implementation of a sustainable certificate can help to redefine infrastructure and create sustainable hospitality buildings according to climate change needs and the SDGs of the 2030 Agenda.

## 2. Materials and Methods

In this conceptual framework, the study was designed in three levels:

The first level was to select the relevant case studies. In order to compare and examine different buildings of the same typology, these case studies needed to be based in the UK, present different construction stages and present different BREEAM scores.

The second level consisted of a qualitative research to provide an in-depth understanding of how these BREEAM scores were achieved and what design changes were needed in the process. In order to achieve this, two different groups of four people each were interviewed about the selected case studies in which they were involved in. All of the interviewees were professionals specialized in hospitality, such as architects, designers and technicians.

The third level was the selection and analysis of the non-numerical data of each of the case studies.

Nowadays, BREEAM is globally recognized and applied throughout the world, but it originated in England, where it is the most used sustainability assessment and certification process in the country. As such, all of the hotels in this paper were based in the United Kingdom. The selected case studies were either finalized or under construction. They also have different final BREEAM scores or, in the case of the ones under construction, different BREEAM goals to be achieved.

This qualitative research was utilized to understand "how" and "why" the changes were applied in these hotels. The approach based on personal interviews with technicians was considered to be the best alternative due to its flexible condition. Data analysis focused on retaining rich meaning of the case studies instead of numerical data. Due to the COVID-19 restrictions, interviews were conducted through Yealink Meeting and recorded with Windows 10 Screen Recorder. The survey was done in May 2020, and a total of six hotels were selected as case studies. For privacy reasons, the names of the projects have been replaced by A, B, C, D, E and F. The semi-structured interviews were designed with a selection of open-ended questions based on previous literature reviews [12,14,18–20]. All case studies comprised the questions shown in Table 1, but as the conversation progressed, some further related questions were added with the aim to collect as much deep information as possible for each hotel. In total, fifteen main questions were asked and divided in three categories: design and planning, BREEAM and sustainable design changes.

*Data Collected*

Table 2 shows a summary of each case study including the stage, number of rooms, number of floors, BREEAM goal and BREEAM final score of each hotel. BREEAM considers ten categories to measure the sustainability of the building: Energy, Health and Well-being, Land Use and Ecology, Management, Materials, Transport, Water, Waste, Pollution and Innovation. Each one of them frames different requirements that can be fulfilled according to the chosen strategy. A BREEAM advisor will determine when these requirements are obtained and will score them; these points undergo an environmental weighting factor that classifies them as Outstanding, Excellent, Very Good, Good and Pass [22].

**Table 1.** Main Questions.

| Category | Question |
|---|---|
| Design and planning | How many floors and total number of rooms does this hotel have? |
| | Is this hotel a new building or a restored building? |
| | Is the building finished or under construction? |
| | Does it have planning permission? |
| | Is there any specific planning requirement for this hotel? |
| BREEAM | What was the BREEAM goal score for this building? |
| | Did the BREEAM goal score match the BREEAM final score? |
| | Was BREEAM certification part of the planning requirements? |
| | Was the BREEAM advisor considered during the design stage of the hotel? |
| | Which ones were the main BREEAM categories for this building? |
| | What were main design changes involved in the building to obtain the BREEAM score? |
| Sustainable design changes | Is there any low-consumption technology used in this site? |
| | Was the ecology of the building relevant in this case? |
| | Did the envelope of the hotel change? |
| | Was the landscape of the building redesigned? |

**Table 2.** Hotels Case Studies.

| Case Studies | Stage | No. of Rooms | No. of Floors | BREEAM Goal Score | BREEAM Final Score |
|---|---|---|---|---|---|
| A | Finalized | 216 | 5 | Excellent | Excellent |
| B | Finalized | 302 | 7 | Very Good | Very Good |
| C | Under Construction | 82 | 9 | Very Good | In progress |
| D | Finalized | 339 | 42 | Excellent | Outstanding |
| E | Under Construction | 456 | 13 | Excellent | In progress |
| F | Under Construction | 329 | 19 | Very Good | In progress |

Hereafter, all case studies are discussed to identify the most relevant aspects that have changed in their designs during all construction stages to make a positive impact in their BREEAM score.

1.  Case study A. For this hotel, the objective was to obtain BREEAM Excellent. The score was successfully achieved and reached the maximum score obtained by a hotel at the time of its construction. The building used low-consumption technology to reduce its energy demand and to generate its own energy supply, reducing its $CO_2$ emissions by 87%. It collected rainwater that was used for bathrooms and to irrigate the outdoor garden areas. Finally, it was necessary during the design stage to increase the size of the bicycle parking to promote this practice.

2.  Case study B was a rehabilitation and extension of a historic building with the planning requirement of achieving a Very Good BREEAM score. The architects increased the cycling area. Under the advice of the ecologist and due to the demolition of the adjacent building, they included a series of shelter boxes for bats on the roofs so that they could nest in them. Regarding the building envelope, some windows had to be upgraded as a result of BREEAM acoustic requirements.

3.  Case Study C was a remodel and extension of an existing building and was being built on site at the time of the interview. It aimed to obtain a Very Good BREEAM score. In this case, there was not enough space for landscaping or green surfaces. To compensate for this absence, the presence of mechanical and electrical engineers was crucial. The installation of flow limiters helped save water by reducing the output in

taps and showers. Additionally, technicians had to optimize the existing glazing by adding double-pane windows.

4. Case study D was a sustainable skyscraper with 42 floors and a small footprint. It achieved a score of Outstanding; the ecology category was the main contributing factor for that score. Solar panels and a green roof were included, as well as beehives on the 39th floor that produced honey, which was then consumed by guests. Shelter boxes for bats and birds were also incorporated into the lower levels. Due to its small size on the ground floor, a separated parking area from the building was needed to store the bicycles. This one was designed with a green roof and walls. The mechanical system in this hotel was also of vital importance. It included a thermal energy system that reduces $CO_2$ emissions by 30% and a light regulation system that adjusts depending on the day and season. Furthermore, the hotel recycled cooking oil to produce its soaps and bath products.

5. Case Study E was approved and under construction at the time of the interview. They submitted a preliminary report that established a potential score of 79% for an Excellent BREEAM score. In this case, it was also necessary to increase the size of the bicycle parking and the recycling point. For the ecological requirements, the architects included 65 square meters of greenery distributed in three roof levels. Inside, they increased the thickness of the partitions. They also had to improve the glazing specifications on some of the external windows. Finally, a study of adaptability was necessary to demonstrate how the hotel can change its use if necessary. This was possible due to the structure being based on blade columns and light-weight partitions.

6. Case study F is currently under construction and has a planning requirement to obtain a Very Good BREEAM score. To achieve this, the main changes made by the technicians thus far include an increase in the recycling and bicycle area and the inclusion of planters on the ground floor. Currently, the exterior lighting, including the hotel logo, have been pointed out by the BREEAM advisor to be taken into account. Therefore, an expert lighting technician will undertake a study to present the best solution and reduce light pollution.

## 3. Results and Descriptive Data Analysis

During the design process of the case studies, several changes have been key in achieving BREEAM accreditation under the instruction of the BREEAM assessor. These improvements have been classified into three main groups: layout, performance and additions.

### 3.1. Upgrades through Layout

These changes include internal and external modifications to the design. The increase in the area of the bicycle facilities is a common factor in all of the hotels. This change belongs to the Transportation section of the certificate that promotes exercise and helps reduce $CO_2$ emissions. These credits were given due to the adequate provision of bicycle parking spaces. In some of the hotels, it was also necessary to increase the recycling areas, such as case studies A, B and E. This change belongs to the Waste section; BREEAM aims to promote sustainable waste management and divert items that are recyclable from the landfill or incinerator. It has been shown that around 30% of a hotel's solid waste can be recycled and reused [23].

### 3.2. Upgrades through Performance

Another category of sustainable measures applied especially in rehabilitated and existing buildings are those related to performance. The window typology was changed by updating the glazing specifications and increasing the thickness of the partitions to meet the acoustic requirements of BREEAM. These changes are associated with the acoustic performance subcategory of the Health and Wellness section of BREEAM and were necessary in case studies B, C and E. In this same section, a proposal to reduce light emissions and

ensure that artificial lighting is examined at the design stage to minimize contamination, as it causes degradation in ecosystems.

### 3.3. Upgrades through Additions

Finally, there are specific elements that will help in achieving a green hotel, which can be classified into two main categories, namely, Ecology and Energy. Certain actions are taken to optimize the ecology of the site, such as shelter boxes for birds and bats and flower boxes located in strategic places in the hotels, which occurred in case studies B, D and E. Moreover, including green roofs improves air and water quality and creates habitat for wildlife. In the category of Energy, solar panels and cooling systems can contribute to the reduction of energy consumption in hotels and are an immediate response that can be included to an existing building to optimize its sustainability [24]. BREEAM promotes the reduction of air pollution and carbon emissions and encourages the production of local energy from renewable sources. Renewable energy plays an important role in reducing the effects of climate change and global warming. Accurate research of renewable energy power is crucial in the completion of the 2030 Agenda [25].

These findings demonstrate that the measures to be taken are common in almost all hotels. For rehabilitated buildings, the most important categories to consider are Health and Well-being and Land Use and Ecology. In the case of new hotels, the categories Waste and Transportation have been a common factor. In addition to the elements mentioned above, the incorporation of green roofs and the inclusion of new technological measures, such as solar panels or solar cooling systems, are also key in new hotels [24].

In the case of hotels, it has been proven that the main categories in which BREEAM has the greatest impact are Health and Well-being, Land Use, Ecology and Waste and Transportation. Therefore, if we compare these categories with the SDGs that BREEAM and the UNWTO indicate, we can affirm that the certificate applied in hotels contributes directly to the achievement of SDG 3, 6, 7, 12 and 15 and indirectly in SDG 8, 11 and 13. Moreover, it has been verified that BREEAM supports the inclusion of new technological measures, such as solar panels, which is why it also favors SDG 9. On the contrary, it has not been possible to verify that BREEAM supports SDG 14 in the case of hotels. It should be noted that none of the buildings studied here are near the coast, therefore the sustainable use of marine resources through the sustainable management of fisheries and aquaculture, indicated in SDG 14, is not feasible, but it may be in a coast hotel.

## 4. Discussion

The current study provides a number of interesting findings for hoteliers, design experts and academics. The influence that sustainable certificates have on climate change has been widely studied. Specifically for tourism, Zeppel and Beaumont investigated the $CO_2$ actions by environmentally certified tourism businesses [26]. Their research revealed that hoteliers and operators implemented actions in water, energy and waste reduction and were aware of the consequences of climate change for tourism. This study confirms the evident relationship between climate change and sustainable certificates. It also confirms our finding implying the significance of BREEAM on hotels with regard to SDG 13 [27].

The study conducted by Potrč Obrecht et al. evaluates three different building certification schemes, LEED, BREEAM and DGNB, and analyses their coverage of the main health and well-being aspects in buildings [28]. Their results show that issues such as comfort, light or air quality that affect the inhabitant are addressed in all of the certification assessments. These results provide evidence that BREEAM contributes to the SDG 3 [27]. Furthermore, Haroglu revealed the importance of operational energy, water consumption and materials to ensure a high BREEAM rating [20]. These findings corroborate the direct relation of the assessment with SDG 6 to "ensure availability and sustainable management of water and sanitation for all", SDG 7 to "ensure access to affordable, reliable, sustainable and modern energy for all" and SDG 12 to "ensure sustainable consumption and production patterns" [27]. Lamy et al. also studied the potential contribution and benefits that

green certificates might have on urban sustainability [29]. Their work demonstrates that when environmental certifications are applied to a high number of buildings in the city, it results in energy savings of 9.9 million MWh and enough water savings for a month and a half of water supply.

SDG 15 states: "protect, restore and promote sustainable use of terrestrial ecosystems, sustainably manage forests, combat desertification, and halt and reverse land degradation and halt biodiversity loss" [27]. Pedro et al. [30] highlight the importance of urban planning tools to enhance sustainable use of land and propose a combination of geographical information systems with BREEAM-Communities as a valuable tool for urban planning. Their model confirmed the significance of land use and ecology to BREEAM. It also suggests an advanced tool that can help with the achievement of SDG 15. Their findings also provide evidence of the positive impact BREEAM might have on the city. Similarly, SDG 11 hopes to "make cities and human settlements inclusive, safe, resilient and sustainable" [27]. Additionally, the study carried out by Hamedani and Huber [31] concluded that sustainable certificates can measure and guarantee sustainable development achievements in any region by adopting adequate strategies. Their findings are extremely useful for the 2030 Agenda, as they demonstrate that sustainable assessments, such as BREEAM or LEED, can be applied in any region of the world, and they can also be used by various groups if the right criteria and objectives are identified.

Finally, Norouzi and Soori have studied the ecological, social and economic aspects that assessment methods take into account when evaluating the sustainability of a building [32]. Their research proved that BREEAM has the highest rank for the economic criterion of all the analyzed certificates, with 24.1% of the total score. This finding supports SDG 8, which promotes inclusive and sustainable economic growth [27].

## 5. Conclusions

This study represents a major advancement in the success of the 2030 Agenda and reflects the impact that tourism and the hotel sector have in its achievement. We provide initial insights into the benefits that the BREEAM energy certificate, applied in hotels, may offer to successfully accomplish the SDGs. We found that the certificate contributes to the achievement of 9 of the 17 SDGs proposed by the 2030 Agenda. Nevertheless, the contribution to SDG 14, proposed by the UNWTO, has not been demonstrated through the case studies presented here since this objective focuses on unviable marine resources at our hotels. Despite the absence of studies regarding the impact of sustainable hotels on achieving the goals of the 2030 Agenda, this article demonstrates that an adequate touristic and hospitality approach can generate promising and substantial results in accomplishing SDGs. The current research allows the hospitality industry to better position itself and its role in the sustainable tourism field in the 2030 Agenda; however, it presents some limitations. The case studies employed in this article were not able to confirm any contribution to eight of the SDGs, including: 1, No poverty; 2, Zero hunger; 4, Quality education; 5, Gender equality; 10, Reduced inequalities; 14, Life below water; 16, Peace, justice and strong institutions; and 17, Partnerships for the goals. Some of these goals may not be related to the hospitality industry and contribution to their achievement may not be possible. On the other hand, others, such as goals 5 and 10, can be addressed by the hospitality industry and could make a positive impact on the 2030 Agenda. Hence, further research focused on all types of sustainable hotels, including coastal hotels, and their contributions will be necessary to delve into this field and provide more data pertaining to which SDGs are most influenced by the sustainable hotel industry. Additionally, further research examining the reduction of energy consumption for rehabilitated buildings will help hoteliers improve their existing hotels in a more sustainable approach. Finally, data of social perception towards green hotels will be beneficial in confronting global climate change and contribute to sustainable tourism.

Hotels play a major role in sustainable tourism. The current paper reveals the adaptations that hotels can make to obtain a high BREEAM score, which contributes to a number

of SDGs. This innovative approach demonstrates the use of a BREEAM certificate as a tool for hoteliers and hospitality stakeholders to directly and indirectly fulfill 9 of the 17 SDGs and place them in a favorable position to confront the 2030 Agenda. Further research is needed to explore how hotels may contribute to the unexplored SDGs of this study.

**Author Contributions:** Conceptualization, M.M.S.-B., R.E.H.F. and P.T.-T.; methodology, M.M.S.-B.; software, M.M.S.-B.; validation, M.M.S.-B., R.E.H.F., P.C.-C. and P.T.-T.; formal analysis, M.M.S.-B.; investigation, M.M.S.-B., R.E.H.F. and P.T.-T.; resources, M.M.S.-B., R.E.H.F. and P.T.-T.; data curation, M.M.S.-B.; writing—original draft preparation, M.M.S.-B.; writing—review and editing, M.M.S.-B.; visualization, M.M.S.-B.; supervision, R.E.H.F. and P.T.-T.; project administration, R.E.H.F.; funding acquisition, R.E.H.F. All authors have read and agreed to the published version of the manuscript.

**Funding:** This work was supported for publication by the "Conselleria de Innovación, Universidades, Ciencia y Sociedad Digital" of the Generalitat Valenciana. The authors would like to also thank the INGEGRAF association.

**Conflicts of Interest:** The authors declare no conflict of interest.

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
