# Peer review of "How the Implementation of BREEAM in Hotels Could Help to Achieve the SDGs"

_applsci, doi:10.3390/app112311131_

Round 1
Reviewer 1 Report
This study has some important messages to the sustainable development. However, it is doubted that whether the research method has properly been chosen. it is suggested that other than interviewing hotel professionals, a quantitative research should be undertaken to prove the relationship between BREEM and achievement of SDGs. The results cannot substantiate the discussion and conclusions as it is merely a summary of the operational changes of the construction of hotels. We have no ideas on what data were collected during the interviews and how they are analyzed.
Reviewer 2 Report
This paper is short but informative. I enjoyed reading it and I would like to give few minor suggestions on how to improve it. Since all chapters are very short, I suggest that you modify your paper by deleting subchapters. If you need to highlight improvements in Results, you could use bullets or numbers.
Methodology
How many people were interviewed? Were there more interviewed people than case studies?
Please, provide information about data analysis.
Conclusions
Along with future research suggestions you should give limitations of your research. Additionally, you should also clearly state the contributions of your research (scientific, managerial etc.)
Reviewer 3 Report
The main idea of the article is excellent, but the overall merit is really low.
Scientific soundness of the article is absolutely low for this kind of journal.
Could authors describe or highlight more and widely what is their contribution, their innovative approach and what can be applied from their work? The article seems now to be a literature review.
I am sure that after revisions could be the article reviewd again and has a potential to be published.
Round 2
Reviewer 1 Report
The comments have been addressed for publication.Reviewer 3 Report
Authors improved the article really a lot. The current state of an article is ready to be published in this form.
Articles shows now what was needed.